# Donkey and Hybrid Anaesthetic Mortality in an Observational, Prospective, Multicentre Cohort Study

**DOI:** 10.3390/ani15131880

**Published:** 2025-06-25

**Authors:** Hannah Boocock, Jana Flyps, André Escobar, José I. Redondo, Polly M. Taylor, Miguel Gozalo-Marcilla, G. Mark Johnston, Regula Bettschart-Wolfensberger, Rebekah Sullivan

**Affiliations:** 1Veterinary Department, The Donkey Sanctuary, Exeter EX14 9SU, UK; 2Department of Large Animal Surgery, Anaesthesia and Orthopaedics, University of Ghent, Salisburylaan 133, 9820 Merelbeke, Belgium; jana.flyps@ugent.be; 3Department of Clinical Sciences, Ross University School of Veterinary Medicine, Basseterre P.O. Box 334, Saint Kitts and Nevis; aescobarvet@gmail.com; 4Departamento de Medicina y Cirugía Animal, Facultad de Veterinaria, Universidad Cardenal Herrera-CEU, CEU Universities, 46115 Valencia, Spain; nacho@uchceu.es; 5Taylor Monroe, Little Downham, Cambridgeshire CB6 2TY, UK; 6The Royal (Dick) School of Veterinary Studies and The Roslin Institute, The University of Edinburgh, Easter Bush Campus, Edinburgh EH25 9RG, UK; miguelgozalomarcilla@gmail.com; 7Vetstream Ltd., Three Hills Farm, Bartlow, Cambridge CB21 4EN, UK; mark.johnston@vetstream.com; 8Department of Clinical Diagnostics and Services, Vetsuisse Faculty, University of Zürich, 8057 Zürich, Switzerland; rbettschart@vetclinics.uzh.ch; 9Society for Protection of Animals Abroad, 55 Ludgate Hill, London EC4M 7JW, UK; rebekah.sullivan@spana.org

**Keywords:** donkey, hybrid, hinny, mule, anaesthesia, standing sedation, mortality, CEPEF4

## Abstract

Anaesthetic mortality in donkeys and donkey–horse hybrids has not previously been specifically examined. The aim of this worldwide observational, prospective, multicentre cohort study was to report on mortality in the 7 days following a general anaesthetic or standing sedation in donkeys and hybrids. Data were collected as part of the fourth Confidential Enquiry into Perioperative Equine Fatalities (CEPEF4). A total of 825 cases were included, with 757 donkeys and 68 hybrids. Of these cases, 616 donkeys and 56 hybrids underwent a general anaesthetic. The overall mortality rate for general anaesthesia in donkeys in this study was 1.0% and was higher in hybrids at 3.6%. There were 141 standing sedations performed in donkeys and 12 in hybrids. Mortality rates for standing sedations were lower, with the donkey mortality rate of 0.7%, and no hybrid mortality was seen in the study. Donkey mortality appears to show similar trends to those of the horse, whilst hybrid mortality appears to be higher. However, the numbers are too low to draw definitive conclusions. Further details around the general anaesthesia and standing sedation of donkeys and hybrids are described in this paper.

## 1. Introduction

Anaesthetic mortality in equids has long been of interest to equine practitioners, owing to the higher mortality rates associated with equine anaesthesia compared to other species, such as dogs and cats [1,2,3]. The Confidential Enquiry into Perioperative Equine Fatalities 2 (CEPEF2) included data regarding donkeys and mules [1]. Despite this, to date, there has not been a large-scale study specifically investigating the perioperative anaesthetic mortality risk in donkeys and donkey–horse hybrids (mules and hinnies).

The true global population of donkeys and hybrids is not known with certainty, but was estimated at around 52.9 million donkeys and 7.7 million hybrids [4]. Many of these are found in low- and middle-income countries as working equids [5]. These species are also often kept as non-working companion animals in the USA, the UK, and other European countries. Despite this large global population, familiarity and confidence with performing anaesthesia in these species are often lacking amongst veterinary practitioners. Whilst there is some published evidence of key differences to horses, the literature in these species related to sedation, anaesthesia, morbidity, and mortality is scarce. Furthermore, practitioners may not have extensive experience with donkey and hybrid handling and management. Behaviour, anatomy, physiology, and drug metabolism may impact the perioperative assessment, management, and delivery of safe sedation and anaesthesia [6,7].

Donkeys and hybrid behaviour differ substantially from horses, especially with regard to pain [7,8,9]. Donkeys are stoic, which risks underestimation or misinterpretation of any pain, stress, or severity of illness [9]. Hybrids may display a mixed behavioural repertoire between that of a donkey and horse, which may vary depending on the parent breed and prior experiences with human interaction [7,10]. Hybrids can be challenging to handle and sedate, if fearful.

Anatomy can also present a challenge. The donkey’s thick skin without a clear jugular groove can make the placement of an intravenous catheter more difficult. For general anaesthesia (GA), intubation of the trachea can also be challenging due to the large pharyngeal recess and comparatively small trachea [11]. However, the smaller size of most donkeys may contribute to a lower risk of myopathy/neuropathy [12].

Further key differences include physiology. Physiological variables in donkeys are described and are different from those of horses, such as faster heart and respiratory rates and lower core temperature [11]. The mixed species parentage of hybrids means variables such as heart rate, respiratory rate, and temperature are likely to differ from their parent species, but these variables are poorly described in the literature. This may make pre-anaesthetic assessment of the patient more challenging, thereby affecting anaesthetic risk [11].

Finally, drug metabolism in donkeys and hybrids is often different from horses. Donkeys have a higher cellular water content and a faster metabolic rate than horses, with further variation in hepatic and renal metabolism—all of which will contribute to differences in drug dosing, efficacy, and safety [13,14]. Examples include the variable metabolism of non-steroidal anti-inflammatory drugs (NSAIDs), such as phenylbutazone, flunixin meglumine, carprofen, and meloxicam [15,16,17,18], and the shorter half-lives of ketamine and guaifenesin [19,20]. Variability in response to anaesthetic regimes has been seen between horses, mules, and donkeys when using a standard protocol [21,22,23], suggesting that the aforementioned variability in drug metabolism and behaviour is relevant to anaesthesia in donkeys and hybrids.

Data on anaesthetic mortality and risk in donkeys and hybrids were collected as part of the worldwide observational, prospective, multicentre cohort fourth Multicentre Confidential Enquiry into Perioperative Equine Fatalities (CEPEF4). The main objective of this paper was (i) to report on mortality in the 7 days following a GA or standing sedation in donkeys and hybrids. Also, (ii) to report the different protocols, techniques, and monitoring used for GA and standing sedation, along with descriptive data for the patient and procedure. We hypothesised that (i) donkey and hybrid mortality following anaesthesia will be higher than that of horses [2], owing to the aforementioned differences. Also, that (ii) similar protocols to those used for the anaesthesia of horses [2] will be used in the anaesthesia of donkeys and hybrids, such as drug combinations, monitoring, and recovery protocols.

## 2. Materials and Methods

Donkey- and hybrid-specific data were collected as part of the CEPEF4 study and followed the same protocol as described therein [3,24]. Ethical approval was granted by the Ethical Committee of the Association of Veterinary Anaesthetists (AVA) (protocol 2020-009).

The inclusion criteria were any donkey or hybrid undergoing GA or standing sedation requiring at least one ‘top up’ or a constant rate infusion (CRI) to maintain sedation. Procedures, including surgery or advanced diagnostic imaging (e.g., computed tomography scan or magnetic resonance imaging), were included. Procedures where no surgery occurred or only basic imaging occurred (radiographs, sinoscopy, etc.) were excluded. Cases were collected from all participating locations that saw cases fitting the criteria

The full process for data collection and cleaning is described in Gozalo-Marcilla, M., et al., 2025 [2] (Appendix A). Following the same study design as CEPEF2 [1], CEPEF4 implemented information technology (IT) improvements [24]. In brief, data were initially collected using a portable document format (PDF) questionnaire sent via email, but this evolved towards a website with an error management system; this system minimised data inaccuracies and improved the overall efficiency and ease of use of a large dataset [24].

Two hundred and forty-nine variables were included in the questionnaire, including signalment of the animal, the specifics of each GA or standing sedation, and the outcome of the procedure within 7 days (ALIVE, DEAD, or EUTHANASIA). The data were checked for discrepancies and errors and validated for each centre using a 12-question checklist (Appendix A) by MGM (main investigator) and the ambassador of each centre (see Appendix B). Finally, using the Microsoft interface on the Azure website, six structured query languages (SQLs) (Q1–Q6) were executed to address some remaining inconsistencies and allowed for manual revision and correction of affected cases (SQLs pseudo-code in Appendix C). Regarding the outcomes, MGM classified the fatalities throughout the study. After data collection and cleaning with each centre (see Appendix C), each fatality was assessed and classified independently by RBW and PMT. In case of discrepancies, GMJ acted as a referee before data analysis.

Procedures were classified as COLIC or NON-COLIC, using the same criteria as in the main CEPEF4 paper [2]. Animals were classified as ALIVE if alive at 7 days, EUTHANASIA, or DEAD. If the equid died or was euthanised due to an inoperable lesion or died because of pre-existing disease, the case was classed as EUTHANASIA. If death was unexpected or euthanasia was due to complications during the procedure, the case was classed as DEAD.

The data were collected concurrently with the horse-specific data during CEPEF4 [2], with the same categorisation. Each individual user classified and included the information from each case during data entry. Signalment included the species (or donkey–horse hybrid) (HORSE, PONY, DONKEY, and MULE), sex (GELDING, NON-PREGNANT FEMALE, PREGNANT FEMALE, and STALLION), age (NEONATE (<one month), FOAL (one to 12 months), YOUNG (≥1–5 years), ADULT (≥5–14 years), and OLD (>14 years)), body condition score (BCS) (THIN for 1/3, AVERAGE for 2/3 and FAT for 3/3), and physical status based on the American Society of Anesthesiologists (ASA) classification.

The reason for the procedure was recorded. Some animals had multiple procedures performed during the same GA/standing sedation. Reasons for surgery were classified as COLIC for an emergency abdominal surgery, e.g., for intestinal surgery for colic or other abdominal emergencies such as caesarean section, bladder repair, or NON-COLIC. The NON-COLIC included ABDOMINAL for any other non-emergency laparotomy, DIAGNOSTIC for diagnostic purposes, e.g., imaging, ENT for ear/nose/throat procedures, FRACTURE for limb fractures only, ORTHOPAEDIC for other orthopaedic procedures, e.g., keratoma removal, UROGENITAL for procedures including the urinary tract and genitalia, e.g., castration, and, finally, MISCELLANEOUS for other reasons, e.g., sarcoid removal and dental extraction.

Descriptive information about the GA/standing sedation was also collected. The procedure was classified as SCHEDULED for pre-arranged elective procedures, NON-SCHEDULED for elective procedures not scheduled beforehand, and URGENT for procedures performed without being scheduled, as it is an emergency. Procedures were classed as NORMAL if performed within normal working hours or OUT OF HOURS if performed outside normal working hours. The duration of GA/standing sedation was collected in minutes and then classified for analyses as <1 h, 1–2 h, 2–3 h, and >3 h. The type of GA/standing sedation was classified as INH for inhalation only (*including IV top-ups excluding CRIs), PIVA for partial intravenous anaesthesia, e.g., donkey being maintained on an inhalant anaesthetic also being given a lidocaine CRI, TIVA for total intravenous anaesthesia, and STANDING for standing procedures. Information on medications used for premedication, induction (GA only), maintenance, including CRIs/PIVA, and recovery was collected. Combinations of additional medications were noted when used. The induction of GA was recorded as FREE for induction without assistance or head restraint only, GATE for a gate-assisted induction, PERSONNEL ASSISTED if supported by 2 or more people against the wall, and SLING if a sling or TABLE if a tilt table were used. Monitoring equipment used during the procedure was collected on a YES/NO basis, as was the use of locoregional anaesthesia and the use of a ventilator. Finally, the recovery method was recorded as FREE for an unassisted recovery, MANUAL if assisted by 1 or more people, ROPES if assisted using ropes, and SLING if assisted using another method, such as a sling or pool. Recovery quality was scored between 1–5, with 1 being excellent and 5 being very poor.

Descriptive statistical analysis was performed on the data. Donkey and hybrid data are presented separately due to species differences. No further statistical analysis was performed due to the small sample size.

The STROBE-Vet guidelines, an extension of the STROBE (Strengthening the Reporting of Observational Studies in Epidemiology) statement [25], were followed (https://www.sciencedirect.com/science/article/pii/S0167587716303282?via%3Dihub, accessed on 1 May 2025) (Appendix A). The STROBE-Vet guidelines set out the methods and processes of developing and strengthening the reporting of observational studies in epidemiology in order to maximise reporting quality.

## 3. Results

Overall, 63 centres from 22 countries collaborated in the systematic collection of data on donkeys and donkey–horse hybrids. Owing to the ongoing recruitment process, each centre commenced data collection on a different date, during the period from November 2020 to June 2023. Figure 1 presents a heat map that illustrates the spatial distribution of documented cases of donkeys and hybrids. After data cleaning, 825 cases were verified, comprising 757 donkeys and 68 hybrids (Figure 2).

Among the donkey cases, 616 required GA, whereas 141 were managed with standing sedation. Only 28 of these cases were conducted in the field, which include 17 TIVAs, one inhalation anaesthesia, and 10 standing sedation procedures.

Among the hybrid cases, 56 required GA, whereas 12 were managed with standing sedation. Only 7 of these cases were conducted in the field, all of them TIVAs. Demographic data of donkeys and hybrids are shown in Table 1.

### 3.1. Mortality

#### 3.1.1. General Anaesthesia

Shown in Table 2, Table 3 and Table 4, the mortality analysis revealed that 6 of the 616 donkeys undergoing GA were classified as DEAD, representing a mortality rate of 1.0%. Among these, four deaths were classified as NON-COLIC DEAD cases, accounting for 0.7%, while two deaths were classified as COLIC DEAD cases, with a mortality rate of 5.3%. Among the 56 hybrids that underwent GA, 2 died, yielding a mortality rate of 3.6%. Both fatalities were categorised as NON-COLIC, corresponding to 4.3%. One death occurred in recovery and one death occurred one day after the GA.

Euthanasia was reported in 22 of the donkeys undergoing GA, representing 3.6% of the total. Of these, there were 14 NON-COLIC cases (2.4%) and 8 COLIC cases (21.1%). Four hybrids who underwent GA were EUTHANASIAs, representing 7.1%. Three cases were classified as COLICs; all occurred during maintenance, with a rate of 33.3%, while one case was NON-COLIC, accounting for 2.1%, which occurred three days post GA.

#### 3.1.2. Standing Sedation

For standing sedations (see Table 2), 1 of the 141 donkeys died, corresponding to a mortality rate of 0.7%, and was classified as a NON-COLIC DEAD. This donkey died on day 3 after the standing sedation. For standing sedations, none of the 12 hybrids died.

For standing sedations, there were two EUTHANASIA cases, representing 1.4%. Both cases were associated with NON-COLIC surgeries; one donkey was euthanised during the maintenance of the sedation, the other was euthanised one day after the standing sedation. No hybrids were euthanised during standing sedation procedures.

### 3.2. General Anaesthesia (Drugs and Protocols, and Monitoring)

Figure 3 and Figure 4 detail the drugs given during each phase of GA. Table 5 describes the drugs and combinations used for premedication. Table 6 and Table 7 detail induction methods and drugs given. Table 7, Table 8 and Table 9 describe maintenance agents, including CRIs. Figure 5 and Figure 6 illustrate how the donkeys and hybrids undergoing GA were monitored. Table 10, Table 11 and Table 12 cover drugs given for recovery, methods of recovery, and recovery quality scores.

### 3.3. Standing Sedation (Drugs, Protocols, and Monitoring)

From the 63 collaborating centres, 17 submitted standing sedation cases of donkeys and 6 submitted standing sedation cases of hybrids. The drugs and protocols for the standings and sedations are shown in Figure 7 and Figure 8 and Table 13. Table 14 reports the different CRIs used to maintain standing sedation.

During standing sedation, ECG was monitored in 8 (6.1%) donkeys, SpO_2_ in 3 (2.3%) cases, and temperature in 71 (50.4%) cases. Non-invasive blood pressure was monitored in 7 (5%) cases. No other type of monitoring was performed during standing sedations.

During standing sedation, ECG was monitored in 4 of the 12 hybrids (33.3%), SpO_2_ in 2 (16.7%), temperature in 4 cases (33.3%), and non-invasive blood pressure was monitored in 2 (16.7%). No other type of monitoring was performed during standing sedations.

## 4. Discussion

The results of this study report do not support our initial hypothesis: the mortality for donkeys undergoing GA was not higher than horses, actually being a similar 1.0% compared with 1.2% in horses in CEPEF4 [2]. Hybrids undergoing GA had a much higher mortality rate of 3.6%. However, the numbers are too low in this study to draw definitive conclusions; further investigation with increased numbers is required. Our second hypothesis was partially supported, as similar drug protocols were used in donkeys and hybrids compared to horses. Slightly less monitoring was performed in donkeys compared to horses, and recovery protocols were also different [2]. Monitoring and recovery protocols used in hybrids were more similar to horses than donkeys, but again, the dataset for hybrids was too small to form a definitive conclusion.

### 4.1. Mortality Data

Whereas the overall death rate in CEPEF4 for horses and ponies undergoing GA was 1.2%*,* COLICs 4.2%, and NON-COLICs 0.6% [2], these numbers were 1.0%, 5.3%, and 0.7%, respectively, for donkeys undergoing GA. Although comparisons should be performed carefully due to the different number of cases (47,396 horses and ponies vs. 616 donkeys undergoing GA), this dataset is of value as it is the first worldwide observational, prospective, multicentre cohort collecting cases reporting information for these species. There were insufficient numbers of hybrids to make sound conclusions, with only 56 hybrids undergoing GA in this data pool. From the initial results, the percentage mortality for hybrids may be higher (overall 3.6%*,* COLICs 0.0%, and NON-COLICs 4.3%). Future studies will increase sample sizes, providing more meaningful results.

Focusing on the data of donkey GAs, one potential reason for the slightly lower overall death rate in donkeys vs. horses (1.0% vs. 1.2%) could be the study population. Many procedures had fewer risk factors for mortality, with most donkeys undergoing short, routine, elective surgeries, with low ASA grades [1,2]. The recoveries appear to be calmer, with most having a recovery score of 1 or 2 [26,27]. There were few neonatal donkeys (<1%) undergoing GA; this may reflect either a lower incidence of surgical conditions in neonates or, more likely, the tendency for these cases to have high perioperative mortality or not be referred for surgery due to perceived poor prognosis or lower economic value [28,29]. Additionally, horses are more likely to be bred intentionally than donkeys, thereby reducing the potential frequency of a donkey neonate requiring surgery. These data may also reflect the demographics of the study population, with centres contributing a different number of cases and potentially having differing populations. This introduces a potential bias, whereby high-risk neonates are underrepresented, possibly skewing mortality percentages towards a more favourable outcome. A significant proportion of procedures were for miscellaneous conditions (classified as NON-COLIC), such as the treatment of sarcoids and urogenital conditions, such as castration. Both groups represented almost 70% of the donkey study population. These are often short, minimally invasive surgeries in otherwise healthy animals, again potentially contributing to the low overall mortality observed [27]. The same pattern is observed in the ASA grading, with 90% of the donkeys having an ASA I or ASA II grade. The majority of surgeries (84.4%) were under two hours in duration; shorter anaesthetic times are associated with improved survival in horses and probably in donkeys as well [1,26,27].

Colic surgeries in the donkey, while relatively few in number, had a notably higher rate of intraoperative death (5.3% COLIC vs. 0.7% NON-COLIC) or euthanasia (21.1% COLIC vs. 2.4% NON-COLIC), consistent with the existing literature in horses, where colic surgery has higher mortality [1,2,30]. This subgroup often represented emergency cases; animals presented for colic surgery may have had a worse prognosis at presentation or be euthanised intraoperatively on welfare or financial grounds. Furthermore, outcomes for surgical colics in donkeys have been noted to be relatively poor, with one study reporting a survival to discharge rate of 54.5% [31]. Colic requiring surgery in the donkey may be under-recognised by clinicians due to the stoic nature of the donkey [9]. The non-specific clinical signs compared to horses may lead to a delay in surgical treatment and potentially result in poorer anaesthetic and surgical outcomes due to the severity of the disease. Further study into anaesthetic mortality associated with colic in donkeys is thus warranted.

Standing surgeries, which avoid GA, had a very low associated mortality rate; however, the numbers are too small to draw meaningful conclusions. Only 3 out of 141 donkeys undergoing standing procedures died or were euthanised, supporting trends in the equine literature suggesting lower mortality risk with standing surgery [2].

### 4.2. General Anaesthesia (Drugs and Protocols)

A wide variety of drugs was used to perform GA and standing sedation in the study, reflecting protocols used with horses [2], supporting our second hypothesis. This variety of drugs and combinations used demonstrates that multiple approaches can be used for premedication and anaesthetic induction in donkeys and hybrids. The choice of the agent may be made depending on the indication for anaesthesia of the donkey or hybrid, clinician preferences, availability and cost of medications, and familiarity with a particular protocol.

Premedication in donkeys was commonly achieved with combinations of an alpha-2-agonist and a full or partial agonist–antagonist opioid (34.8% of cases) or alpha-2 and opioid with the addition of acepromazine (48.2% of cases). Hybrids were more often given alpha-2 alone (32.1%) or alpha-2 with a full or partial agonist–antagonist opioid (35.7%) compared to donkeys. The reason for reduced use of ACP in hybrids is unclear but may reflect individual clinician preference, as acepromazine is frequently used in horse and donkey anaesthesia [2].

The induction of GA was widely achieved with ketamine and benzodiazepine combinations (85.7% of donkeys, 73.2% of hybrids), similar percentages to the 84.6% reported in horses [2]. Other combinations were used less frequently, including ketamine-propofol or tiletamine-zolazepam. Maintenance of GA varied between donkeys and hybrids. Donkeys were more likely to be maintained on inhalant agents alone, most commonly isoflurane (55.4% INH vs. 27.9% PIVA vs. 16.7% TIVA). As in horses [2] and in contrast to donkeys, hybrids were more likely to receive PIVA with isoflurane and an alpha-2-agonist CRI (50.0% PIVA vs. 19.6% INH vs. 30.4% TIVA). However, PIVA is still less commonly used in hybrids compared to horses [2]. The lower employment of PIVA in donkeys and hybrids may reflect a lack of clinician familiarity with PIVA in these species, the often short duration of surgery, and the potential underperception of pain in donkeys. One centre most often used inhalation anaesthesia alone, so may also represent variability in other factors such as anaesthetic case load or management practices. Limited use of PIVA may also be related to concerns about the differing metabolism of drugs commonly used in PIVA in horses. There are reports of suspected lidocaine toxicity in a donkey [32], and ketamine metabolism in donkeys differs from horses and between different breeds of donkeys [19,21]. Protocols used in horses may need to be adapted to the donkey. Further research is needed to demonstrate the anaesthetic effects of PIVA in donkeys and hybrids, which appear to be beneficial for minimising the cardiovascular effects of GA and improving recovery in horses [33]. However, most donkeys in the study appeared to have good quality recoveries, which may demonstrate reduced need for PIVA to improve recovery quality in donkeys due to their calmer, stoic nature or for short, elective procedures where analgesia can be sufficiently provided by other means, such as locoregional anaesthesia.

Locoregional techniques were used in 43.2% of the donkey GAs and 21.4% of hybrid GAs vs. only 18.4% in the horse dataset [2]. The caseload of the study is likely to have influenced this figure, with a high percentage of miscellaneous procedures, such as sarcoid removals, castrations, and orthopaedic procedures, wherein local anaesthesia techniques can be easily implemented. The increased use of locoregional techniques in donkeys and hybrids represents a positive variation from horses. Attention should be paid to the NSAIDs used in donkeys and hybrids for painful procedures, as some variation in NSAID metabolism is shown between species. Meloxicam has been demonstrated to have a very short half-life in donkeys, with a questionable analgesic effect [18]. Donkeys also metabolise flunixin faster than horses, whilst hybrids appear to metabolise flunixin at a similar rate to horses [16]. Pain scores for donkeys have been validated and used clinically [8,9], which may provide useful information for the pre- and post-operative management of donkey pain. No pain scores exist for hybrids. Pain has been associated with poor recovery in horses [26], and adequate pain management may be related to the generally good quality of recovery seen in donkeys and hybrids in this study, alongside other factors such as the donkey’s behavioural tendency towards stoicism.

The majority of donkeys and hybrids had an uneventful, calm recovery, with scores of predominantly 1 or 2 (about 90% in donkeys and about 80% in hybrids) (see Table 12). Compared with horses, in which around two-thirds received a small dose of an alpha-2-agonist prior to or for recovery, only about one-fifth of the donkeys received an alpha-2-agonist alone or in combination [2]. Previous studies have shown poor quality recovery in donkeys induced with propofol without premedication and maintained on isoflurane alone [34]. Isoflurane alone was a commonly used maintenance agent in this study population. However, the use of balanced premedication combinations and induction with ketamine benzodiazepine mixtures frequently used herein may be related to the better quality of recovery seen, as maintenance with isoflurane alone does not appear to have adversely affected recovery quality. These findings also support the clinical impression that donkeys tend to recover from GA in a calmer, more controlled manner than horses, particularly in elective cases. Donkeys have a behavioural tendency for caution due to their strong sense of self-preservation, as opposed to the horse’s greater tendency towards flight [7]. This, together with their generally smaller body weight (Table 1), facilitates manually assisted recovery (35.7%) or recovery without assistance (60.1%). In this study, rope-assisted recovery in donkeys was rarely used (3.8%).

Conversely, the high percentage of donkeys receiving no pharmacological support during recovery raises the question of whether discomfort or anxiety may be under-recognised in donkeys. Given the known stoicism of donkeys and their subtle pain and stress-related behaviour, it is possible that pain or distress during recovery is underestimated. However, it may be that as many donkeys in this study were anaesthetised for short periods for elective procedures, additional medications above what has been given in the premedication and intra-operatively may be unnecessary. Clinicians undertaking donkey anaesthesia should familiarise themselves with pain recognition and anaesthetic management in donkeys and take these into consideration when making decisions about analgesia and recovery [6,8,9].

In hybrids, sedation before or during recovery was more common, with 42.9% receiving some sedation, predominantly alpha-2-agonists alone (37.5%). The higher rate of sedative use in hybrids may reflect behavioural unpredictability during recovery or increased clinician caution due to uncertainty surrounding hybrid recovery patterns and may reflect a sometimes larger body size, making manual assistance more challenging (Table 1). Hybrids are often considered less predictable and more reactive than donkeys, lacking the characteristic stoicism that can make donkeys easier to manage [7]. As a result, clinicians may be more inclined to use sedatives or analgesics during recovery to minimise risk and maintain safety for both the animal and handlers. Despite the increased use of sedation, a notable proportion of hybrids still recovered without intensive assistance. Free recovery was performed in 32.1% and 24.5% were manually guided. However, 41.5% underwent rope-assisted recovery, a higher proportion than reported in donkeys, possibly reflecting greater concerns about control or stability during recovery. Rope recovery may also be more feasible in hybrids due to their larger body size and sufficient tail hair to attach ropes to. These data suggest that their more unpredictable recovery behaviour may prompt more cautious handling strategies.

### 4.3. Monitoring

Monitoring during GA in donkeys and hybrids differed from use in the horse [2]. The majority of cases were monitored with an ECG (electrocardiogram), SpO_2_ (peripheral saturation of haemoglobin with oxygen by pulse-oximetry), and EtCO_2_ (partial pressure of end-tidal carbon dioxide) in similar percentages as in horses. However, donkeys were much less likely to have invasive blood pressure monitored than horses. Placement of intravenous catheters has been noted to be more challenging in donkeys, and arterial catheters may be equally difficult to place. Arterial blood gas (ABG) analysis is also less frequently performed in donkeys compared to horses overall, probably largely due to lack of arterial access, but also potentially due to the smaller number of colic surgeries performed in the sample population. COLICs appeared to be monitored more intensively than NON-COLICs, including more monitoring of arterial blood pressure (78.9% vs. 39.8%, respectively) and ABG (65.8% vs. 35.8%), probably due to the higher risk associated with colic surgeries.

### 4.4. Standing Sedation (Drugs, Protocols and Monitoring)

For procedures performed under standing sedation, premedication was most commonly performed using detomidine and butorphanol combinations, with most animals either maintained on a detomidine CRI or boluses of detomidine and/or butorphanol. Donkeys and hybrids were much more likely to be given local anaesthesia for standing sedation (83.7% of donkeys and 75.0% of hybrids) than horses [2]. Fewer diagnostic procedures were performed in the study population when compared to horses in CEPEF4 [2], but this alone does not account for the increased use of locoregional anaesthesia in donkeys. Monitoring of donkeys was limited, with temperature being measured in approximately 50% of cases, which was higher than that described in CEPEF4 [2]. The low mortality rate combined with relatively minimal machine-based monitoring may again indicate a case selection bias but may also support the potential for standing sedation to be safer than general anaesthesia.

### 4.5. Limitations

Our study has some limitations. First, the number of individuals is too low for statistical analysis and thereby to extrapolate and make comparisons with larger datasets such as the CEPEF4 dataset [2]. Future studies to collect a larger sample size are indicated. Second, the study may be biased by the centre. Slightly more than 50% of the cases were sent by 3 centres of the 63 that reported cases of donkeys and hybrids, which may have influenced the results. Finally, most of the population reported here consisted primarily of non-working equids, with experienced and well-equipped clinicians to manage cases in medium- to high-income countries. Whilst this is beneficial for comparison to horses from the original CEPEF studies [1,2], these results should be extrapolated with caution to global donkeys and hybrids. Most of the world’s donkeys and hybrids are working animals, often in resource-poor locations and treated in centres with a probable different caseload and availability of drugs and equipment. This report may not be an accurate representation of donkey and hybrid anaesthesia in all settings, and further study is required to evaluate donkey and hybrid anaesthesia mortality in different settings.

## 5. Conclusions

Donkeys appear to have a similar overall mortality rate for general anaesthesia to horses, using similar techniques and anaesthetic protocols as those that are described for horses. Hybrid mortality appears to be higher, but a small dataset should indicate caution with this conclusion. Colic mortality in donkeys appears to be higher compared to horses, potentially due to the donkey’s stoic nature and different behavioural repertoire. Further study with a larger sample size and a more diverse study population would be beneficial for examining specific risk factors for mortality in donkeys and hybrids.

## Figures and Tables

**Figure 1 animals-15-01880-f001:**
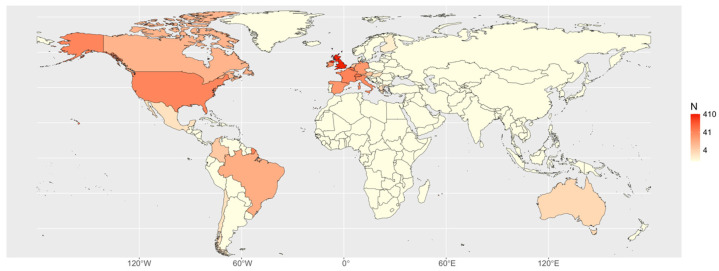
Heat map of the distribution of the cases by country.

**Figure 2 animals-15-01880-f002:**
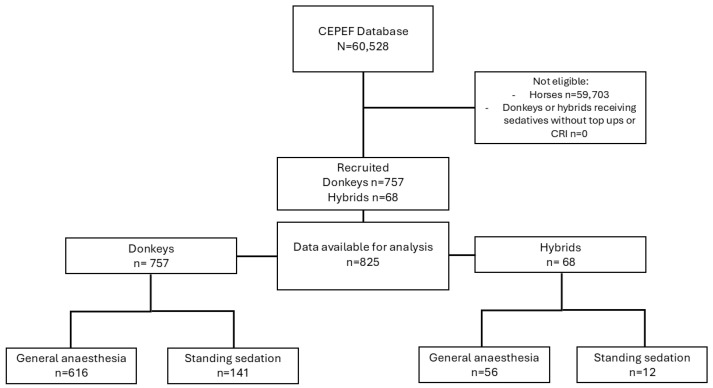
Flow diagram describing the cases included or excluded in the study.

**Figure 3 animals-15-01880-f003:**
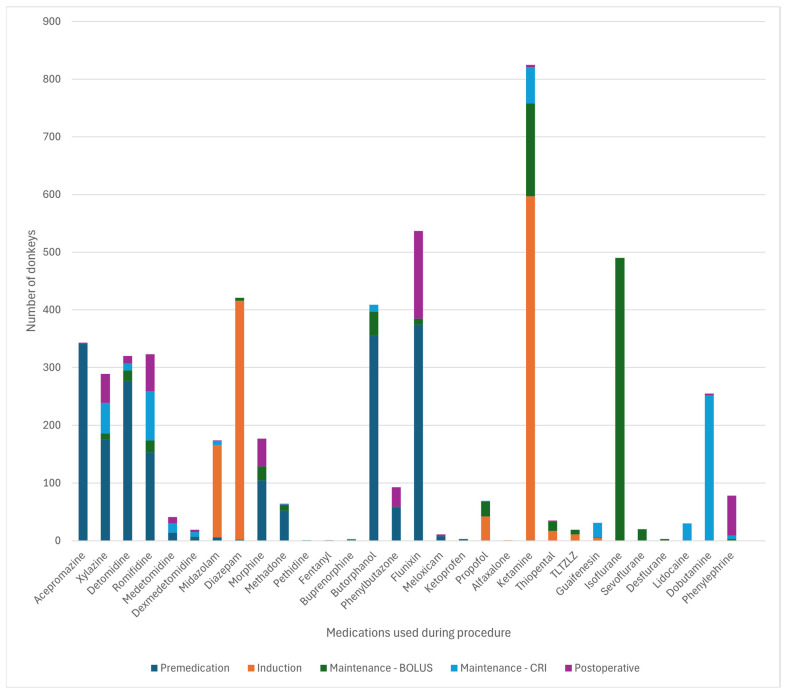
Frequency of the drugs used at each phase of general anaesthesia in 616 donkeys. TLTZLZ: tiletamine-zolazepam. CRI—constant rate infusion.

**Figure 4 animals-15-01880-f004:**
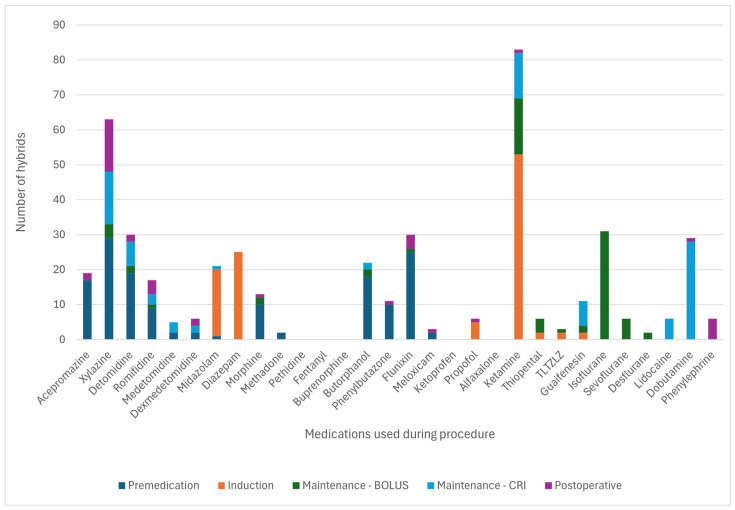
Frequency of the drugs used at each phase of general anaesthesia in 56 hybrids. TLTZLZ: tiletamine-zolazepam; CRI: constant rate infusion.

**Figure 5 animals-15-01880-f005:**
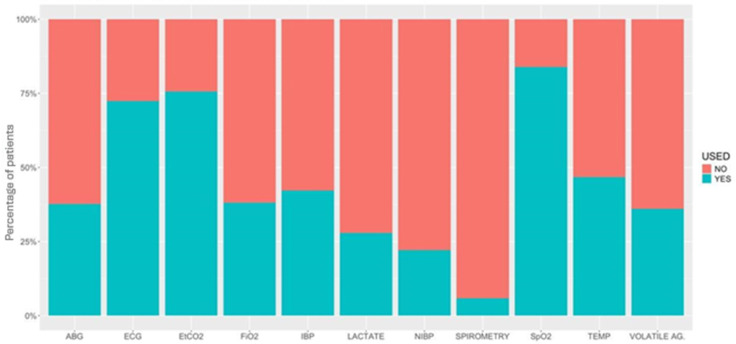
Percentage of patients under different types of monitoring used in 616 donkeys undergoing general anaesthesia. ABG, arterial blood gases; ECG, electrocardiogram; EtCO_2_, partial pressure of end-tidal carbon dioxide; FiO2, inspiratory fraction of oxygen; IBP, invasive blood pressure; NIBP, non-invasive blood pressure; SpO_2_, partial saturation of haemoglobin with oxygen by pulse-oximetry; TEMP, temperature; VOLATILE AG., inspired fraction and end-tidal concentration of inhalant anaesthetics.

**Figure 6 animals-15-01880-f006:**
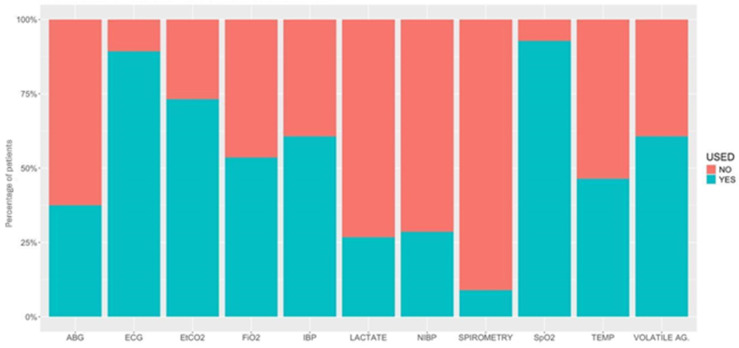
Percentage of patients under different types of monitoring used in 56 hybrids undergoing general anaesthesia. ABG, arterial blood gases; ECG, electrocardiogram; EtCO_2_, partial pressure of end-tidal carbon dioxide; FiO_2_, inspiratory fraction of oxygen; IBP, invasive blood pressure; NIBP, non-invasive blood pressure; SpO_2_, peripheral saturation of haemoglobin with oxygen by pulse-oximetry; TEMP, temperature; VOLATILE AG., inspired fraction and end-tidal concentration of inhalant anaesthetics.

**Figure 7 animals-15-01880-f007:**
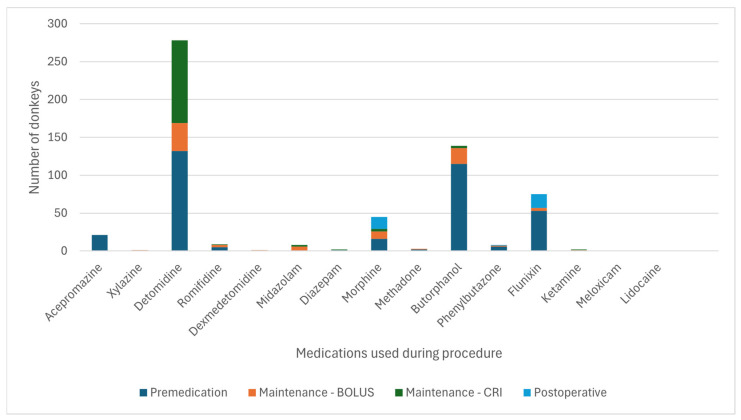
Frequency of the drugs used at each phase for standing sedation in 141 donkeys. CRI: constant rate infusion.

**Figure 8 animals-15-01880-f008:**
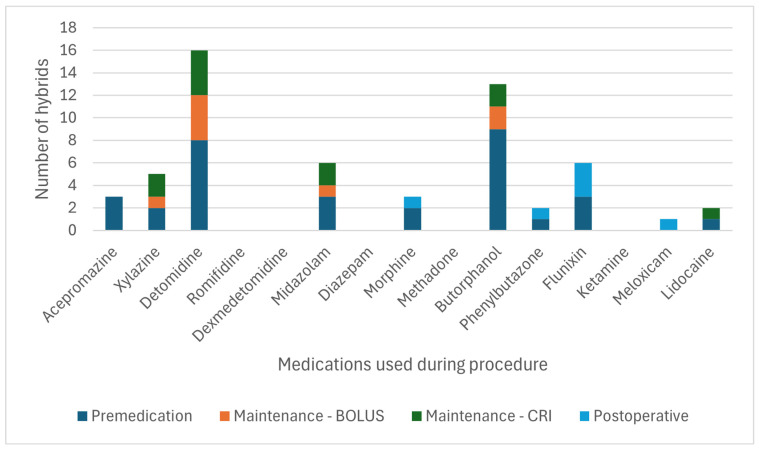
Frequency of the drugs used at each phase for standing sedation in 12 hybrids. CRI: constant rate infusion.

**Table 1 animals-15-01880-t001:** Demographic data by general anaesthesia or standing sedation and overall in 757 donkeys and 68 hybrids.

		General Anaesthesia		Standing Sedation		Overall	
		Donkeys (*n* = 616)	Hybrids (*n* = 56)	Donkeys (*n* = 141)	Hybrids (*n* = 12)	Donkeys (*n* = 757)	Hybrids (*n* = 68)
Sex	Gelding	263 (42.7%)	21 (37.5%)	62 (44.0%)	4 (33.3%)	325 (42.9%)	25 (36.8%)
	Stallion	215 (34.9%)	15 (26.8%)	16 (11.3%)	4 (33.3%)	231 (30.5%)	19 (27.9%)
	Non-preg. female	129 (20.9%)	20 (35.7%)	63 (44.7%)	4 (33.3%)	192 (25.4%)	24 (35.3%)
	Pregnant female	9 (1.5%)	0 (0%)	0 (0%)	0 (0%)	9 (1.2%)	0 (0%)
Age *	Neonate	5 (0.82%)	1 (1.8%)	1 (0.7%)	2 (16.7%)	6 (0.80%)	3 (4.4%)
	Foal	66 (10.8%)	5 (8.9%)	1 (0.7%)	0 (0%)	67 (8.9%)	5 (7.4%)
	Young	138 (22.7%)	15 (26.8%)	4 (2.8%)	3 (25.0%)	142 (18.9%)	18 (26.5%)
	Adult	273 (44.8%)	21 (37.5%)	42 (29.8%)	6 (50.0%)	315 (42.0%)	27 (39.7%)
	Geriatric	127 (20.9%)	14 (25.0%)	93 (66.0%)	1 (8.3%)	220 (29.3%)	15 (22.1%)
BCS	Normal	504 (81.8%)	44 (78.6%)	126 (89.4%)	12 (100%)	630 (83.2%)	56 (82.4%)
	Fat	69 (11.2%)	6 (10.7%)	8 (5.7%)	0 (0%)	77 (10.2%)	6 (8.8%)
	Thin	43 (7.0%)	6 (10.7%)	7 (5.0%)	0 (0%)	50 (6.6%)	6 (8.8%)
Weight (KG)	Mean ± SD	182 ± 78	339 ± 214	182 ± 58	370 ± 166	182 ± 75	365 ± 174
ASA	I	396 (64.3%)	21 (37.5%)	91 (64.5%)	4 (33.3%)	487 (64.3%)	25 (36.8%)
	II	154 (25.0%)	18 (32.1%)	42 (29.8%)	7 (58.3%)	196 (25.9%)	25 (36.8%)
	III	34 (5.5%)	8 (14.3%)	8 (5.7%)	0 (0%)	42 (5.5%)	8 (11.8%)
	IV	22 (3.6%)	4 (7.1%)	0 (0%)	0 (0%)	22 (2.9%)	4 (5.9%)
	V	10 (1.6%)	5 (8.9%)	0 (0%)	1 (8.3%)	10 (1.3%)	6 (8.8%)
Reason for anaesthesia ^1^	Abdominal	19 (3.1%)	3 (5.4%)	1 (0.7%)	0 (0%)	20 (2.6%)	3 (4.4%)
	Colic	38 (6.2%)	9 (16.1%)	0 (0%)	0 (0%)	38 (5.0%)	9 (13.2%)
	Diagnostic	49 (8.0%)	3 (5.4%)	7 (5.0%)	2 (16.7%)	56 (7.4%)	5 (7.4%)
	ENT	5 (0.8%)	0 (0%)	5 (3.5%)	0 (0%)	10 (1.3%)	0 (0%)
	Fracture	3 (0.5%)	0 (0%)	0 (0%)	0 (0%)	3 (0.40%)	0 (0%)
	Miscellaneous	205 (33.3%)	14 (25.0%)	129 (91.5%)	10 (83.3%)	334 (44.1%)	24 (35.3%)
	Orthopaedic	110 (17.9%)	16 (28.6%)	2 (1.4%)	0 (0%)	112 (14.8%)	16 (23.5%)
	Urogenital	212 (34.4%)	14 (25.0%)	2 (1.4%)	0 (0%)	214 (28.3%)	14 (20.6%)
Colic surgery	Colic	38 (6.2%)	9 (16.1%)	0 (0%)	0 (0%)	38 (5.0%)	9 (13.2%)
	Non-colic	578 (93.8%)	47 (83.9%)	141 (100%)	12 (100%)	719 (95.0%)	59 (86.8%)
Type of anaesthesia	Inhalation	341 (55.4%)	11 (19.6%)	0 (0%)	0 (0%)	341 (45.0%)	11 (16.2%)
	PIVA	172 (27.9%)	28 (50.0%)	0 (0%)	0 (0%)	172 (22.7%)	28 (41.2%)
	Standing	0 (0%)	0 (0%)	141 (100%)	12 (100%)	141 (18.6%)	12 (17.6%)
	TIVA	103 (16.7%)	17 (30.4%)	0 (0%)	0 (0%)	103 (13.6%)	17 (25.0%)
Duration *	<1 h	279 (45.4%)	20 (35.7%)	50 (35.7%)	3 (27.3%)	329 (43.6%)	23 (34.3%)
	1–2 h	241 (39.1%)	18 (32.1%)	64 (45.7%)	7 (63.6%)	305 (40.5%)	25 (37.3%)
	2–3 h	71 (11.6%)	12 (21.4%)	19 (13.6%)	0 (0%)	90 (11.9%)	12 (17.9%)
	>3 h	23 (3.7%)	6 (10.7%)	7 (5.0%)	1 (9.1%)	30 (4.0%)	7 (10.5%)
Locoregional	No	350 (56.8%)	44 (78.6%)	23 (16.3%)	3 (25.0%)	373 (49.3%)	47 (69.1%)
	Yes	266 (43.2%)	12 (21.4%)	118 (83.7%)	9 (75.0%)	384 (50.7%)	21 (30.9%)
Ventilation	No	435 (70.6%)	24 (42.9%)	141 (100%)	12 (100%)	576 (76.1%)	36 (52.9%)
	Yes	181 (29.4%)	32 (57.1%)	0 (0%)	0 (0%)	181 (23.9%)	32 (47.1%)
Timetable	Normal	574 (93.2%)	45 (80.4%)	140 (99.3%)	12 (100%)	714 (94.3%)	57 (83.8%)
	Out of hours	42 (6.8%)	11 (19.6%)	1 (0.71%)	0 (0%)	43 (5.7%)	11 (16.2%)
Scheduled	Non-scheduled	25 (4.1%)	3 (5.4%)	6 (4.3%)	0 (0%)	31 (4.1%)	3 (4.4%)
	Scheduled	520 (84.4%)	37 (66.1%)	125 (88.7%)	10 (83.3%)	645 (85.2%)	47 (69.1%)
	Urgent	71 (11.5%)	16 (28.6%)	10 (7.1%)	2 (16.7%)	81 (10.7%)	18 (26.5%)

^1^ Donkeys and hybrids could be anaesthetised/sedated for more than one reason. ASA, American Society of Anaesthesiologists; BCS, body condition score; ENT, ear–nose–throat; KG, kilogram; PIVA, partial intravenous anaesthesia; TIVA, total intravenous anaesthesia; * Missing data for category age of 7 donkeys in GA (donkeys in GA: *n* = 609, total donkeys: *n* = 750) and missing data for category duration of 2 donkeys in GA, 1 donkey standing, and 1 hybrid standing (donkeys in GA: *n* = 614, donkeys standing: *n* = 140, total donkeys: *n* = 754, hybrids standing: *n* = 11, total hybrids: *n* = 67).

**Table 2 animals-15-01880-t002:** Mortality rates for donkeys and hybrids under general anaesthesia and standing sedation.

	General Anaesthesia		Standing Sedation		Overall	
	Donkeys (*n* = 616)	Hybrids (*n* = 56)	Donkeys (*n* = 141)	Hybrids (*n* = 12)	Donkeys (*n* = 757)	Hybrids (*n* = 68)
Alive	588 (95.5%)	50 (89.3%)	138 (97.9%)	12 (100%)	726 (95.9%)	62 (91.2%)
Dead	6 (1.0%)	2 (3.6%)	1 (0.7%)	0 (0%)	7 (1.0%)	2 (2.9%)
Euthanised	22 (3.6%)	4 (7.1%)	2 (1.4%)	0 (0%)	24 (3.2%)	4 (5.9%)

**Table 3 animals-15-01880-t003:** Colic versus non-colic mortality rates under general anaesthesia.

	Colic (GA)		Non-Colic (GA)	
	Donkeys (*n* = 38)	Hybrids (*n* = 9)	Donkeys (*n* = 578)	Hybrids (*n* = 47)
Alive	28 (73.7%)	6 (66.7%)	560 (96.9%)	44 (93.6%)
Dead	2 (5.3%)	0 (0%)	4 (0.7%)	2 (4.3%)
Euthanised	8 (21.1%)	3 (33.3%)	14 (2.4%)	1 (2.1%)

**Table 4 animals-15-01880-t004:** Time of death and euthanasia of donkeys under general anaesthesia.

		PREM	IND	MAIN	REC	1D	2D	3D	4D	5D	6D	7D
**Colic**	Deaths (*n* = 2)	0	1	1	0	0	0	0	0	0	0	0
	Euthanised (*n* = 8)	0	0	4	1	1	1	0	0	1	0	0
**Non-colic**	Deaths (*n* = 4)	0	0	1	0	0	1	0	0	1	1	0
	Euthanised (*n* = 14)	0	0	10	0	1	0	1	0	2	0	0
**Total**	Deaths *(n* = 6)	0	1	2	0	0	1	0	0	1	1	0
	Euthanised (*n* = 22)	0	0	14	1	2	1	1	0	3	0	0

**Table 5 animals-15-01880-t005:** Drugs and different drug combinations used for premedication before general anaesthesia in 616 donkeys and 56 hybrids.

Drug or Drug Combinations Premedication	Donkeys (*n* = 616)	Hybrids (*n* = 56)
ACP ^1^ + alpha-2-agonist	43 (7.0%)	8 (14.3%)
ACP + alpha-2-agonist + partial/agonist–antagonist opioid	204 (33.1%)	5 (8.9%)
ACP + alpha-2-agonist + full agonist opioid	93 (15.1%)	3 (5.4%)
ACP + partial/agonist–antagonist opioid	1 (0.2%)	0 (0%)
ACP + alpha-2–agonist + pure opioid + partial/agonist–antagonist opioid	0 (0%)	1 (1.8%)
Alpha-2-agonist + partial/agonist–antagonist opioid	149 (24.2%)	12 (21.4%)
Alpha-2-agonist + full agonist opioid	65 (10.6%)	8 (14.3%)
Alpha-2-agonist alone	55 (8.9%)	18 (32.1%)
Benzodiazepines alone	2 (0.3%)	1 (1.8%)
None	1 (0.2%)	0 (0%)
Partial/agonist–antagonist opioid alone	2 (0.3%)	0 (0%)
Full agonist opioid alone	1 (0.2%)	0 (0%)

^1^ ACP, acepromazine.

**Table 6 animals-15-01880-t006:** Methods of induction of general anaesthesia for 616 donkeys and 56 hybrids.

	Donkeys (*n* = 616)	Hybrids (*n* = 56)
Free	276 (44.8%)	7 (12.5%)
Gate	46 (7.5%)	20 (35.7%)
Personal assisted	293 (47.6%)	29 (51.8%)
Sling	1 (0.2%)	0 (0%)

**Table 7 animals-15-01880-t007:** Drugs used for the induction of general anaesthesia in 616 donkeys and 56 hybrids.

Drug or Drug Combinations Induction	Donkeys (*n* = 616)	Hybrids (*n* = 56)
Alfaxalone	1 (0.2%)	0 (0%)
Benzodiazepine + ketamine	528 (85.7%)	41 (73.2%)
Guaifenesin + ketamine	3 (0.5%)	2 (3.6%)
Ketamine alone	11 (1.8%)	4 (7.1%)
Ketamine + propofol	38 (6.2%)	4 (7.1%)
Propofol	4 (0.7%)	1 (1.8%)
Thiopental + ketamine	17 (2.8%)	2 (3.6%)
Tiletamine + zolazepam	14 (2.3%)	2 (3.6%)

**Table 8 animals-15-01880-t008:** Drugs used for the maintenance of general anaesthesia in 616 donkeys and 56 hybrids.

Drugs Maintenance	Donkeys (*n* = 616)	Hybrids (*n* = 56)
Desflurane	3 (0.5%)	2 (3.6%)
Isoflurane	490 (79.5%)	31 (55.4%)
Sevoflurane	20 (3.2%)	6 (10.7%)
Ketamine	27 (4.4%)	6 (10.7%)
Other ^1^	76 (12.3%)	11 (20.0%)

^1^ IV protocols, such as guaifenesin or midazolam combined with alpha-2-agonists and ketamine.

**Table 9 animals-15-01880-t009:** Drugs used for constant rate infusion (CRI) during general anaesthesia in 616 donkeys and 56 hybrids.

Drugs Constant Rate Infusion (CRI)	Donkeys (*n* = 616)	Hybrids (*n* = 56)
None	417 (67.7%)	21 (37.5%)
Alpha-2-agonist	111 (18.0%)	16 (28.6%)
Lidocaine	13 (2.1%)	2 (3.6%)
Alpha-2-agonist + lidocaine	11 (1.8%)	2 (3.6%)
Ketamine	6 (1.0%)	1 (1.8%)
Alpha-2-agonist + ketamine	39 (6.3%)	10 (17.9%)
Lidocaine + ketamine	6 (1.0%)	2 (3.6%)
Alpha-2-agonist + butorphanol	0 (0%)	2 (3.6%)
Alpha-2-agonist + ketamine + butorphanol	12 (2.0%)	0 (0%)
Alpha-2-agonist + methadone	1 (0.16%)	0 (0%)

**Table 10 animals-15-01880-t010:** Parenteral drugs administered for/during the immediate recovery period after general anaesthesia in 616 donkeys and 56 hybrids.

Drugs Recovery	Donkeys (*n* = 616)	Hybrids (*n* = 56)
ACP + alpha-2-agonist	1 (0.16%)	2 (3.6%)
Alpha-2-agonist + pure opioids	17 (2.8%)	0 (0%)
Alpha-2-agonist alone	124 (20.1%)	21 (37.5%)
None	442 (71.8%)	32 (57.1%)
Pure opioids alone	32 (5.2%)	1 (1.8%)

**Table 11 animals-15-01880-t011:** Methods of recovery from general anaesthesia for 599 donkeys and 53 hybrids.

	Donkeys (*n* = 599)	Hybrids (*n* = 53)
Free	360 (60.1%)	17 (32.1%)
Manual	214 (35.7%)	13 (24.5%)
Ropes	23 (3.8%)	22 (41.5%)
Sling	2 (0.3%)	1 (1.9%)

**Table 12 animals-15-01880-t012:** Recovery scores from general anaesthesia for 596 donkeys and 53 hybrids.

	1	2	3	4	5
Donkeys (*n* = 596) *	380 (63.8%)	151 (23.3%)	47 (7.9%)	13 (2.2%)	5 (0.8%)
Hybrids (*n* = 53)	31 (58.5%)	11 (20.8%)	6 (11.3%)	3 (5.7%)	2 (3.2%)

* Missing data for 2 donkeys. A total of 599 donkeys went into recovery, 2 had missing data, and 1 donkey was euthanised during recovery (*n* = 596). Scoring system classifications: (1) One attempt to stand, no ataxia, (2) One to two attempts to stand, some ataxia, (3) >2 attempts to stand but quiet recovery, (4) >2 attempts to stand, excitation, (5) Severe excitation. Patient injured.

**Table 13 animals-15-01880-t013:** Drugs and different drug combinations used for premedication before standing sedations in 140 donkeys and 12 hybrids. ^1^ ACP, acepromazine. Missing data of 1 donkey during standing sedation (*n* = 140).

Drug or Drug Combinations Premedication	Donkeys (*n* = 140)	Hybrids (*n* = 12)
ACP ^1^ + alpha-2-agonist	2 (1.4%)	0 (0%)
ACP + alpha-2-agonist + partial/agonists-antagonists opioids	6 (4.3%)	1 (8.3%)
ACP + alpha-2-agonist + pure opioids	11 (7.8%)	2 (16.7%)
ACP + partial/agonists-antagonists opioids	1 (0.7%)	0 (0%)
ACP alone	1 (0.7%)	0 (0%)
Alpha-2-agonist + partial/agonists-antagonists opioids	106 (75.2%)	6 (50.0%)
Alpha-2-agonist + pure opioids	7 (5.0%)	0 (0%)
Alpha-2-agonist alone	4 (2.8%)	1 (8.3%)
Partial/agonists-antagonists opioids alone	2 (1.4%)	2 (16.7%)

**Table 14 animals-15-01880-t014:** Drugs and different drug combinations used for constant rate infusion in standing sedations in 141 donkeys and 12 hybrids.

Drugs Constant Rate Infusion (CRI)	Donkeys (*n* = 141)	Hybrids (*n* = 12)
Alpha-2-agonist	104 (73.8%)	3 (25%)
Alpha-2-agonist + butorphanol	3 (2.1%)	2 (16.7%)
Alpha-2-agonist + morphine	3 (2.1%)	0 (0%)
Alpha-2-agonist/lidocaine	0 (0%)	1 (8.3%)
Ketamine	1 (0.7%)	0 (0%)
None	30 (21.3%)	6 (50%)

## Data Availability

Raw data, converted to a .csv file, are stored in the CEPEF CIC metadata file. The most relevant results can be accessed at https://cepef4.wordpress.com/preliminary-results/ (accessed on 1 May 2025). CEPEF CIC is a registered Community Interest Company (CIC), company number 15398252.

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
