# Peer review of "Donkey and Hybrid Anaesthetic Mortality in an Observational, Prospective, Multicentre Cohort Study"

_animals, 2025, doi:10.3390/ani15131880_

Round 1
Reviewer 1 Report
Comments and Suggestions for Authors
This study presents the mortality rate in the 7 days following general anaesthesia and standing sedation in donkeys and hybrids. It also describes the anaesthetic protocols and monitoring used in those species. Data collected within the framework of the CEPEF4 initiative were used. Differences from horses and between donkeys and hybrids are highlighted.
This study shows that overall mortality is higher in hybrids, and similar in donkeys and horses. Those results contradict the authors’ hypothesis according to which donkeys and hybrids would have had higher mortality rates compared to horses.
The study is in line with the objectives, and the manuscript is well written. It adds significant information to the limited body of knowledge regarding non-horse equids anaesthesia.
Please find attached a few comments:
Table 1 pages 6-7:
The small number of hybrids does not allow to draw definitive conclusions regarding mortality associated with general anaesthesia in this species. Even though, mortality rate seems to be higher than in donkeys and horses. Hybrids included in this study present risk factors (more neonates than in donkeys, higher ASA status, more colic surgeries than in donkeys, longer anaesthesia time, more out of hours and urgent procedures than in donkeys) that should be mentioned in the discussion.
Page 7:
L228: 1 donkey died at Day3 after standing sedation. What did it die for? (for my personal information)
Page 14:
L347: “biasing results to only those cases with a better prognosis”. It doesn’t seem to be the case. Indeed, intraoperative mortality rate associated with colic surgery is higher in donkeys (5.3%) compared with horses (4.2%).
Page 15:
L364-366: “ACP is widely used as a tranquilizer for fractious patients as may be more often seen in a hybrid”. This statement is not supported by the results of this study. Indeed, more hybrids are not premedicated with ACP (69.6%) compared with donkeys (44.7%).
L372: According to Table 1 (page 6), 55.4% (and not 67%) of the donkeys are maintained on inhalant agents alone.
L374: Use the same number of decimals for all the results. According to Table 1 (page 6), TIVA in 30.4% (and not 30%) in hybrids.
L387-388: “when considering some aspects of anaesthesia”. This statement needs to be clearer. The only reason that justifies the reduced need for PIVA in donkeys is most likely associated with behavioural differences between horses and donkeys. In my opinion, your statement suggests that other differences may exist (e.g. reduced pain management required in donkeys).
L401-402: “adequate pain management may be related to the generally good quality of recovery seen in donkeys and hybrids in this study”. Again, even if your statement may be correct, I have the feeling that donkeys recover better compared to horses, irrespective of the adequacy of pain management. I would rather stress the point on behavioural differences. Indeed, in my experience, donkeys have a much better quality of recovery, even for very invasive, and potentially painful, procedures.
L402-404: I would move this sentence to 4.3 standing sedation.
L404-405: “this probably reflects the particular need to prevent pain stimulation in a conscious patient”. This statement doesn’t justify why donkeys and hybrids receive better pain management during sedation compared with horses. Standing horses are also conscious animals.
L412-413: “isoflurane alone was a commonly used maintenance agent in this study, but doesn’t appear to have adversely affected recovery quality”. The absence of premedication, and maybe the induction with propofol, are more likely to cause poor recovery quality than maintenance on isoflurane alone reported in [30]. Only 0.2% of donkeys were not premedicated, and only 0.7% were induced with propofol in your study. The protocols described in your study can therefore not be compared with the one used in [30]. Reflecting this comment, I would suggest rethinking/rewording your statement.
Page 16:
L421: I would add somewhere that rope-assisted recovery in donkeys might also be difficult to implement as they don’t have a lot of hair to attach to rope on the tail.
L462-463: “and arterial catheters may be equally difficult to place”. Do you really think so? In my opinion, artery of the ears can easily be used to place an arterial catheter in donkeys.
Page 17:
L473-474: “future studies to collect a larger sample size are indicated”. Based on the number of donkeys recruited in this large multicentric CEPEF4 initiative, over a quite extended period, do you think this is realistic?
L477: “primarily of non-working equids, with experienced and well equipped clinicians to manage cases”. In my opinion, this represents a strength rather than a weakness of the study. Indeed, it means that donkeys treated with the same standards as horses have a similar mortality rate. This result is therefore only species-specific and not environment related. Studying the mortality rate of donkeys in less equipped settings would probably be very interesting but it would not allow to compare those results to the existing CEPEF4 results in horses.
Reviewer 2 Report
Comments and Suggestions for Authors
Thank-you for this submission which adds to our knowledge of mortality in donkeys and hybrid equids. This is mostly clear, well written and analysed. There are some long sentences at times which can be difficult to follow, a few points of terminology and a few inconsistencies which I would like to point out, as well as perhaps some comments on discussion
Line 60-62- a very long sentence, could be subdivided
Line 63: remove hyphen in North America (though North America is a continent not a country?)
Line 64-66: This sentence starts by talking about the confidence of practitioners, but ends by talking about about the published evidence of differences- it would appear to me that knowledge that donkeys are different may well be a factor in lack of confidence. Just because differences are published this does not mean we have familiarity with the actual handling/management of donkeys/hybrids in many centres. Suggest rewording this
Line 72-74: I am hesitant to call mules/hinnies ‘species’ due to their lack of ability to reproduce, in most cases. Can the manuscript maintain use of ‘hydrids’ or similar instead of species, when referring to mules etc?
Line 75: suggest comma “sedate, if fearful”
Line 80: contribute to a
Line 88: please add ‘than horses’
Line 92: Here you use the term guaifenesin, later it is Guaiacol glyceryl ether- I suggest using the same drug name throughout.
Line 104-105 some of these ling sentences with subclauses can be difficult ot follow on a quick read, though being perfectly correct. Perhaps repositioning the subclause to the end of the sentence would help “similar protocols to those used for the anaesthesia of horses will be used in the anaesthesia of donkeys and hybrids (such as drug combinations, monitoring and recovery protocols)”
Line 123: Could you indicate what is meant by ‘cleaned’, more precisely?
Line 137: signalment
Ine 143: Suggest full stop rather than semicolon
Line 162: strictly speaking inhalants you refer to are vapors not gases- but “inhalant” is adequate in my view
Line 167: Should this be personnel assisted rather than personal?
Line 178-182: Please could this part be reworded for clarity. Suggest removing internal subclause and adding to end (e.g. STROBE-vet guidelines were followed in order to maximise reporting quality. Then give part on more info of what these guidelines are)
Figure 3: Some figures have a slightly poor quality in some areas (e.g. drug names in this figure)- perhaps a better quality image would solve this
Figure 4: drug names are much clearer on this figure compared to figure 3
Table 5: agonist-antagonist would be more usual term rather than agonists-antagonists
‘pure opioid’ is not a term I use personally (I appreciate it is common)- I think you are referring to a ‘full’ agonist opioid, and indeed this is the term you use in the discussion later
Table 9: this table feels to be oddly positioned, as information on recovery is placed before the information on method of induction? I appreciate you were placing drug information together, however for me I would prefer information on the phases aligned together.
Figures 5/6: Again could you consider order?
Tables 11/12: It would feel easier to have the sedation for recovery data together with these tables
Line 278: ‘sent’ appears an odd word- would ‘submitted’ or ‘contributed’ work?
Figure 7: again poor quality on the axes
Line 289: here you use the term continuous rate infusion, previously used was constant rate infusion- suggest standardising to one term throughout
Line 320: not sure ‘demographics’ is needed- just state study population
Line 325-327: some repetition here, with the first part stating not referred for surgery, the second part stating centres don’t see them (seems to be the same point again)- perhaps try to reword and combine?
Line 357: suggest separating into 2 sentences by removing semicolon, replace with full stop
Line 362-3: is this comparison with donkeys or horses? Please make clear
Line 404: If I have interpreted your flow diagram correctly, it would appear that none of the donkeys/hybrids had standing sedation for imaging or other prolonged non-surgical reasons. This would appear to mean that 16.3% donkeys and 25% hybrids undergoing standing sedation had surgery WITHOUT local anaesthetics? This appears shocking to me- is it correct? Could you perhaps comment on the acceptability of this (relating to the procedures etc?)?
Line 410: I don’t believe the study referenced was a comparative study, so should this be ‘poor recoveries’; rather than ‘poorer recoveries’ or make clearer that you refer to the difference between your findings and those in that study?
Line 432: could you rephrase to avoid the sentence starting with ’32.1%’
Line 437: I am not sure that ‘support’ is quite the right term?
Line 437-443: I totally accept there is probably under recognition of pain in donkeys, however as the majority of donkeys in this study were anaesthetised for short periods and minor procedures, I would not personally be expecting them to receive or require further opioids for recovery. Could you bring this point in?
Line 445: Although numbers are low for mortality associated with standing sedation, I was actually surprised by how many there were- 3 donkeys dead/euthanased out of 141 feels higher than I expected. I appreciate it is very difficult to draw conclusions on such low numbers, however I wondered if you could maybe comment on this related to the lower figures presented in the CEPEF4 preliminary results in horses?
Line 471: Rather than stating the negative, might be clearer just to state your study has some limitations.
Line 478: ‘should be extrapolated’
Reviewer 3 Report
Comments and Suggestions for Authors
This is an interesting descriptive study that gives information about anesthetic information and associated mortality in donkeys and hybrids. Considering the lack of this type of information, the study adds data for a better understanding in these animals.
It is suggested to change the title about worldwide data, basically because even though it includes 21 countries, it does not reflex the worldwide situation of anesthesia in these species. Maybe authors could consider modifying that part of the title. Also, the word data in title is not necessary.
Why authors consider this is a prospective study? In my opinion, it is a retrospective study. Please explain.
Abstract: the main aim is to report mortality in donkeys and hybrids after GA or standing sedation, but there are other objectives that are included in the results. Please include them here.
Introduction
P2L61: some updated estimated data about donkey and hybrid population is available in FAO webpage (www.fao.org FAOSTAT) that is suggested to be included. E.g. 2023 world population of donkeys is estimated in 52.953.585 and 7.764.602 for mules and hinnies.
P2L63: north America is not a country, please modify.
P2L81-82: it would be interesting if some information about physiological variables is mentioned and how diHer from horses.
P3L100-102: more results were included in aim ii, so please include them here.
Also, general data about animals (sex, age, BCS, weight and ASA) were included in the results, so please also include it as an aim or part of one.
Material and Methods
P3L130-134: the classification of euthanasia or dead was based on previous studies or what was the criteria used to each one? If authors could please explain.
P3L138-140: data was collected from CEPEF4 data, but it would be interesting to have more details about BCS (why was a 1-3 score used and not the 1-5 or 1-9 described). Also, in results (table 1) body weight is shown…how was determined? Estimated by formula (heart girth and height, heart girth and length)? Same for hybrids. Please explain and add the information.
P4L173-174: recovery score from 1 to 5 was the option in the questionnaire? Considering that probably each place uses diHerent score systems. Or data was transformed to homogenize it for these results? Question regarding table 12.
It is true that sample size was small especially in hybrids. However, in the case of donkey data, maybe a logistic regression for mortality risk factors could have been done, with less categories maybe…as a suggestion to be considered.
Results
Considering the small amount of hybrids data that was gathered, maybe the study could be focused only on donkeys. Therefore, more analysis could be done and results (tables and figures) less crowded.
P4L187: 2023..
Figure 1: it is interesting to have this heat map, but authors could include the distribution of the 63 centers by country. This is an important information when results are analyzed considering diHerent realities in each country regarding drugs used, protocols, use of the animals, etc.
P7 Table 4: not sure if needed as a table, maybe this information could be just mentioned in the text.
P7L233: table 2 should be mentioned in this paragraph.
P7L234: Figures 3 and 5…should be 3 and 4
P8 Figure 3 and 4: graphs don’t have Y axis information. Title of both graphs should be frequencies of not list. Also, these figures could be presented al supplement material instead of results and mentioned in the text.
P10 Table 10 this information could also be mentioned in the text and not presented as a table.
P11 Figures 5 and 6: it is suggested to present this as tables for a better understanding of the information.
P12L278 and P17L475: 56 collaborating centers but results in P4L184 it says were 63 centers. Please revise.
P12 Figure 7 an 8. Also consider showing this information as tables.
P17L483: settings.
Discussion
P14L313: even though these results were obtained from 21 countries it is not a worldwide study. Consider change that in the phrase.
Mortality discussion (4.1) almost have no citations, especially the first 2 paragraphs. Please revise and include more previous studies.
P14L357-359: also include something regarding the accessibility to drugs and cost on each center or country, that could be a factor for choosing agents and protocols.
P16L419: some donkeys are smaller than horses, however there are several donkey breeds with bigger heights than horses. Please rewrite this.
P16L426: table 1 is referred in this phrase for body size meaning BCS? If so, it is not correct, so please modify or otherwise explain.
P16L440-442: some information about how pain could be recognized and assessed would be valuable…for example including scores, Grimace scale, etc.
P16L453-455: if standing sedation is safer than GA, what would authors suggest doing, how can clinicians choose?
Limitations
P17L471: suggest deleting the first phrase.
P17L474: results are biased because of this…so, number of centers by country and % of data from each center should be included in the methodology.
It is important to acknowledge that donkeys in these centers are probably pets or do a small amount of work. Also, those centers are located mainly in medium-high income countries, therefore how they are medically treated (including anesthesia and standing sedation) probably is not the reality of most of the donkeys around the world (or where most of the population is located). So, this could be included.
Conclusions: Based on these results, donkeys appear to….
References
Please check the citations because are diHerences, e.g. some journals are abbreviated and some not.
22 is uppercase
Some book don´t have all the information (editor, publisher, pages, etc.)
